# Organ Manifestation and Systematic Organ Screening at the Onset of Inflammatory Rheumatic Diseases

**DOI:** 10.3390/diagnostics12010067

**Published:** 2021-12-29

**Authors:** Tobias Hoffmann, Peter Oelzner, Martin Busch, Marcus Franz, Ulf Teichgräber, Claus Kroegel, Paul Christian Schulze, Gunter Wolf, Alexander Pfeil

**Affiliations:** 1Department of Internal Medicine III, Jena University Hospital, Friedrich Schiller University Jena, Am Klinikum 1, 07747 Jena, Germany; Peter.Oelzner@med.uni-jena.de (P.O.); Martin.Busch@med.uni-jena.de (M.B.); Gunter.Wolf@med.uni-jena.de (G.W.); Alexander.Pfeil@med.uni-jena.de (A.P.); 2Department of Internal Medicine I, Jena University Hospital, Friedrich Schiller University Jena, Am Klinikum 1, 07747 Jena, Germany; Marcus.Franz@med.uni-jena.de (M.F.); Christian.Schulze@med.uni-jena.de (P.C.S.); 3Institute of Diagnostic and Interventional Radiology, Jena University Hospital, Friedrich Schiller University Jena, Am Klinikum 1, 07747 Jena, Germany; Ulf.Teichgraeber@med.uni-jena.de

**Keywords:** inflammatory rheumatic diseases, IRD, connective tissue diseases, newly diagnosed, organ involvement, organ screening

## Abstract

Background: Inflammatory rheumatic diseases (IRD) are often associated with the involvement of various organs. However, data regarding organ manifestation and organ spread are rare. To close this knowledge gap, this cross-sectional study was initiated to evaluate the extent of solid organ manifestations in newly diagnosed IRD patients, and to present a structured systematic organ screening algorithm. Materials and Methods: The study included 84 patients (63 women, 21 men) with newly diagnosed IRD. None of the patients received any rheumatic therapy. All patients underwent a standardised organ screening programme encompassing a basic screening (including lungs, heart, kidneys, and gastrointestinal tract) and an additional systematic screening (nose and throat, central and peripheral nervous system) on the basis of clinical, laboratory, and immunological findings. Results: Represented were patients with connective tissue diseases (CTD) (72.6%), small-vessel vasculitis (16.7%), and myositis (10.7%). In total, 39 participants (46.5%) had one or more organ manifestation(s) (one organ, 29.7%; two organs, 10.7%; ≥three organs, 6.0%). The most frequently involved organs were the lungs (34.5%), heart (11.9%), and kidneys (8.3%). Lastly, a diagnostic algorithm for organ manifestation was applied. Conclusion: One-half of the patients presented with a solid organ involvement at initial diagnosis of IRD. Thus, in contrast to what has been described in the literature, organ manifestations were already present in a high proportion of patients at the time of diagnosis of IRD rather than after several years of disease. Therefore, in IRD patients, systematic organ screening is essential for treatment decisions.

## 1. Introduction

Until a few years ago, rheumatology, on the basis of its historical development, was defined by clinical pictures that mainly represented primary joint disorders such as rheumatoid arthritis (RA), psoriatic arthritis (PsA) or spondyloarthritis (SpA), summarised under the term of rheumatic and musculoskeletal diseases (RMD) [1]. 

With the deeper understanding of immunopathological pathways, advances in diagnostic procedures (e.g., laboratory and imaging), and the introduction of new treatment strategies [1,2], rheumatology underwent a change from RMD to inflammatory rheumatic diseases (IRD) [3]. Additionally, in the last year, new immunomodulatory drugs were developed for the treatment of RMD [4,5,6,7,8] and IRD [9,10,11,12]. 

Given the substantial morbidity and mortality of IRD, early detection and treatment are essential [13,14]. However, its appearance and symptoms are very heterogeneous: IRD can cause a variety of organ involvement presenting with many different and unspecific clinical symptoms [15]. In addition, in some cases, the disease rapidly progresses, making early diagnosis essential also to avoid organ damage [14]. 

However, far fewer than 50% of IRD patients undergo basic organ screening at the time of initial diagnosis [16]. In addition, therapy of IRD has mainly focused on one organ system. For the sake of simplicity, IRD are regarded in studies as isolated single-organ diseases, but as chronic systemic autoimmune diseases, IRD often show multiorgan involvement, which requires a systematic approach.

In this context, there is a substantial knowledge gap regarding the optimal diagnostic procedure of organ involvement in IRD. The objective of the present study was to evaluate organ manifestations (as defined by an involvement of the lungs, heart, kidneys, gastrointestinal tract (GIT), ear, nose, and throat (ENT), and the central and peripheral nervous systems (CNS and PNS)) in IRD patients, and to verify an especially developed organ screening program.

## 2. Materials and Methods

### 2.1. Study Population

We screened 3513 patients ≥ 18 years regarding IRD between October 2015 and December 2019 at the Universal Hospital Jena (Germany), Department of Internal Medicine III. The final analysis encompassed 84 patients with a new onset of IRD (connective tissue disease (CTD): systemic lupus erythematosus (SLE) *n* = 26, systemic sclerosis (SSc) *n* = 15, Sjögren’s syndrome *n* = 15, and mixed connective tissue disease (MTCD) *n* = 4; small-vessel vasculitis: granulomatosis with polyangiitis (GPA): *n* = 5, microscopic polyangiitis (MPA): *n* = 6, eosinophilic granulomatosis with polyangiitis (EGPA): *n* = 3; and myositis: dermatomyositis: *n* = 3, polymyositis *n* = 5, and necrotising myositis *n* = 1). 

None of the patients received any glucocorticoid, immunomodulatory, or immunosuppressive therapy.

### 2.2. Methods

#### 2.2.1. Initial Organ Screening 

The diagnostic process started with a detailed rheumatological medical history focussing on specific or nonspecific symptoms of rheumatic diseases. Pathological laboratory and immunological parameters were also included. This led to the first suspicion of an IRD with potential organ involvement, followed by systematic organ screening. Organ involvement in IRD was defined as IRD manifestation at the following organ systems: lungs, heart, kidneys, GIT, ENT, CNS, and PNS. 

At the beginning, every patient underwent basic organ screening (including lungs, heart, kidneys, and GIT). 

#### 2.2.2. Basic Diagnostic Tests of Systematic Organ Screening

Lungs: pulmonary function tests (PFT), including diffusion capacity measurement (diffusion capacity for carbon monoxide; DLCO);heart: echocardiography;kidneys: quantitative urine protein diagnostic, urine sediment examination, renal ultrasound;GIT (including liver, spleen, intestines): abdominal ultrasound.

Depending on the suspected IRD, additional diagnostic tests were performed, including the evaluation of ENT, CNS, and PNS.

#### 2.2.3. Additional Diagnostic Based on Pathological Laboratory Findings, Clinical Symptoms or Previous Diagnostic:

Lungs: pulmonary high-resolution computed tomography (HRCT), immunological bronchoalveolar lavage (BAL);heart: cardiac magnetic resonance imaging (MRI), myocardial biopsy;kidneys: renal biopsy;GIT (including liver, spleen, intestines): barium swallow, gastroscopy and colonoscopy, biopsy;CNS and PNS: head MRI, neurological examination;ENT: head MRI, ENT examination.

### 2.3. Statistical Analysis

A descriptive statistic was used to evaluate the data. Data were collected in an Excel sheet (Microsoft Windows, Redmond Washington, DC, USA). Lastly, statistical analysis was performed by IBM SPSS Statistics 25 (IBM SPSS Statistics, Chicago, IL, USA, for Windows). *p* < 0.05 was considered to be statistically significant. 

## 3. Results

### 3.1. Baseline Characteristics 

Of the 84 patients, 63 were female (75.0%) and 21 male (25.0%) at a median age of 54.9 ± 15.1 years. The most common IRD was SSc (31.0%), followed by SLE (19.0%), Sjögren’s syndrome (17.9%), MTCD (4.8%), small-vessel vasculitis (16.7% (GPA 6.0%, MPA 7.1%, EGPA 3.6%)), and myositis (10.7% (dermatomyositis 3.6%, polymyositis 5.9%, necrotising myositis 1.2%)) (see Table 2 and Figure 1). 

**Figure 1 diagnostics-12-00067-f001:**
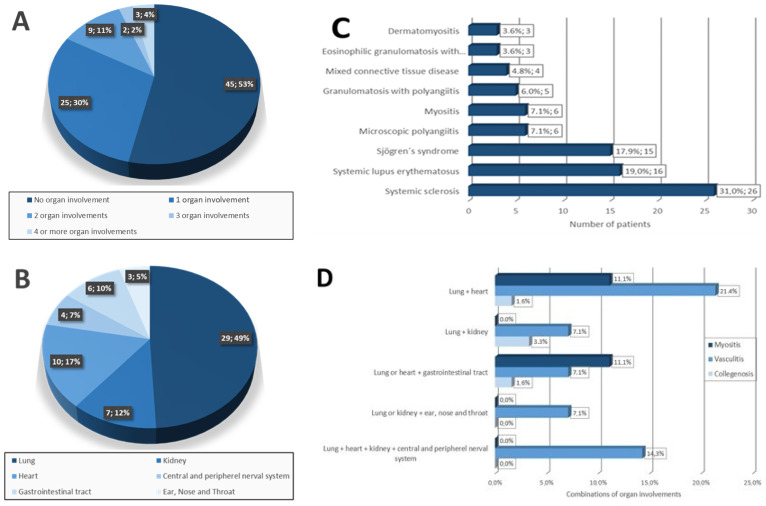
Baseline characteristics. (**A**) Number of organ involvements per patient at initial diagnosis; (**B**) distribution of all affected organs at initial diagnosis, (**C**) distribution of inflammatory rheumatic disease at initial diagnosis, (**D**) distribution of combined organ manifestations in patients with connective tissue disease (CTD), vasculitis, and myositis (CNS/PNS—central and peripheral nerval system; GIT—gastrointestinal tract; ENT—ear, nose, and throat).

### 3.2. Organ Involvement in IRD 

In summary, 39 patients (46.4%) showed the involvement of one or more organs with an average of 1.6 organs affected. Of the study participants, 29.8% had one organ, 10.7% two organs, 2.4% three organs, and 1.2% four or more organ manifestations (see Table 1). Of the 84 patients, 45 (53.6%) presented with no organ involvement. The most frequent manifestation sites were the lungs (*n* = 29, 34.5%), followed by cardiac (*n* = 10, 11.9%), renal (*n* = 7, 8.3%), and GIT (*n* = 6, 7.1%) (see Figure 1 and Table 2).

**Table 1 diagnostics-12-00067-t001:** Number of organ manifestations in IRD patients.

Number of Organ Manifestations	Number	Percentage
None	45	53.6%
1	25	29.7%
2	9	10.7%
3	2	2.4%
≥4	3	3.6%
Total no. of patients with organ manifestation(s)	39	46.4%
Total	84	100.0%

**Table 2 diagnostics-12-00067-t002:** Distribution of diseases and organ manifestation in patients with IRD (gastrointestinal tract—GIT; central and peripheral nervous system—CNS/PNS; ear, nose, and throat—ENT).

	Disease		Number of Affected Organs (Proportionate)	Number of Patients	Percentage (Proportionate)
**Connective tissue disease**	**Systemic sclerosis (SSc)**			26	31.0%
	Organ	Lungs	8 (30.8%)	10	(38.5%)
		GIT	3 (11.5%)		
		Kidneys	2 (7.7%)		
		Heart	1 (3.8%)		
	**Systemic lupus erythematosus (SLE)**			16	19.0%
	Organ	Lungs	2 (12.5%)	6	(37.5%)
		CNS/PNS	2 (12.5%)		
		Kidneys	1 (6.3%)		
		Heart	1 (6.3%)		
	**Sjögren’s syndrome**			15	17.9%
	Organ	Lungs	3 (20.0%)	3	(20.0%)
		Kidneys	1 (7.7%)		
	**Mixed connective tissue disease (MCTD)**			4	4.8%
	Organ	Lungs	1 (25.0%)	2	(50.0%)
		GIT	1 (25.0%)		
**Vasculitis**	**Microscopic polyangiitis (MPA)**			6	7.1%
	Organ	Lungs	5 (83.3%)	5	(83.3%)
		Heart	2 (33.3%)		
		Kidneys	2 (33.3%)		
		CNS/PNS	2 (33.3%)		
	**Granulomatosis with polyangiitis (GPA)**			5	6.0%
	Organ	Lungs	4 (80.0%)	5	(100.0%)
		ENT	3 (60.0%)		
		Heart	1 (20.0%)		
		Kidneys	1 (20.0%)		
		GIT	1 (20.0%)		
		Multiple	1 (20.0%)		
	**Eosinophilic granulomatosis with polyangiitis (EGPA)**			3	3.6%
	Organ	Lungs	2 (66.7%)	3	(100.0%)
		Heart	3 (100.0%)		
**Myositis**	**Myositis**			6	7.1%
	Organ	Heart	2 (33.3%)	2	(33.3%)
		Lungs	1 (16.7%)		
		GIT	1 (16.7%)		
	**Dermatomyositis**			3	3.6%
	Organ	Lungs	3 (100.0%)	3	(100.0%)
	**Total**			84	100.0%

### 3.3. Combination of IRD Organ Involvement 

Organ involvement was most commonly seen in patients with vasculitis, with the highest rate of a combined lung and heart involvement (21.4%). In the presence of CNS/PNS manifestations (14.3%), involvement of the lungs, heart, and kidneys was often simultaneously seen. Patients with CTD most often had involvement of the lungs and kidneys (3.3%), whereas patients with myositis showed a combination of lung and heart involvement (11.1%) (see Figure 1D). 

### 3.4. Basic Diagnostic in Systemic IRD 

#### 3.4.1. Pulmonary Involvement 

PFT, including DLCO, was the first method to be carried out to detect possible lung involvement in 83 patients (98.8%), followed by chest X-ray (*n* = 54, 64.3%). Of 62 patients with pathological findings in the PFT (DLCO < 80%) or in chest X-ray, a pulmonary HRCT was performed in 51 patients, revealing pathological findings in 34 participants (see Figure 3D). Lastly, lung involvement was diagnosed in 29 patients (34.5%), supported by pathological immunological BAL in 21 patients (see Figure 2). 

#### 3.4.2. Cardiac Involvement

All patients received an echocardiography, which showed abnormalities in 22.6% (*n* = 19). Ten out of 22 patients (45.5%) had elevated cardiac troponin I (cTnI) values. On the basis of clinical presentation, echocardiography, and cTnI, the diagnosis of cardiac involvement was verified in four patients. Due to pathological findings in cardiac MRI, cardiac involvement was confirmed in five more patients, and in one patient by myocardial biopsy (see Figure 2 and Figure 3C). In total, ten patients had cardiac involvement.

#### 3.4.3. Renal Involvement

In 78 patients (92.9%), urine protein differentiation was performed, which revealed proteinuria in seven patients (8.3%) (cut-off of ≥ 500 mg/L). In addition, microscopic examination of the urine sediment showed a pathological result in six patients (7.1%). Three participants underwent renal biopsy, confirming renal organ involvement in all of them. In three patients, a biopsy could not be performed due to various contraindications (e.g., dual platelet aggregation inhibition). Lastly, seven patients showed renal manifestation (see Figure 2 and Figure 3E).

#### 3.4.4. GIT Involvement

As part of basic organ screening, 72 patients (85.7%) underwent abdominal sonography, with pathological findings in 4 patients. To evaluate oesophageal involvement in SSc, 12 patients (14.3%) received a barium swallow test, revealing pathological results in 6 patients. In addition, two patients underwent a gastroscopy and colonoscopy, showing GIT manifestations of IRD (see Figure 2).

**Figure 2 diagnostics-12-00067-f002:**
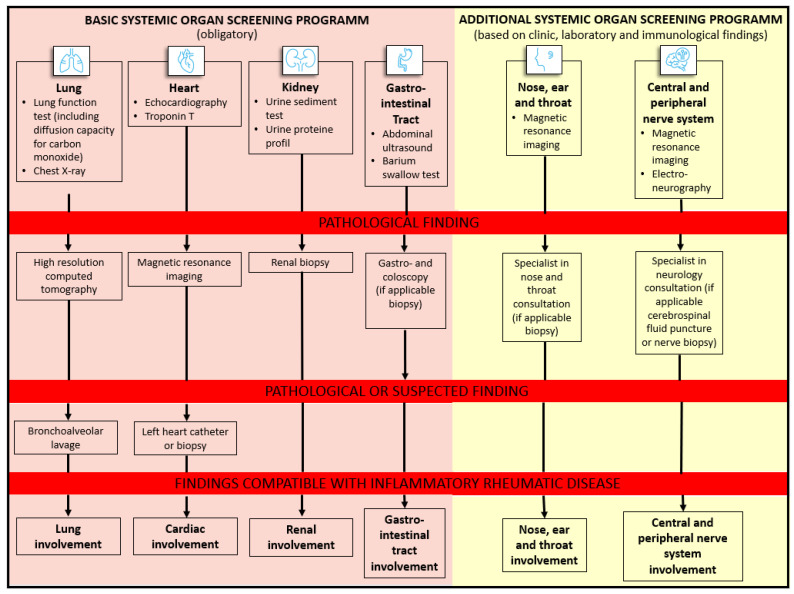
Basic and additional organ screening.

### 3.5. Additional Diagnostics in Systemic IRD

#### 3.5.1. ENT Involvement

ENT involvement was confirmed by head MRI in three patients (3.6%). Two patients were referred to an ENT specialist, but in both cases, no biopsy was necessary (see Figure 2 and Figure 3B).

#### 3.5.2. CNS or PNS Involvement

In 13 patients (15.5%), a head MRI was performed that revealed pathological findings in three cases (23.1% of 12 patients) (see Figure 3A). Seven participants (8.3%) underwent a neurological examination for the further clarification of neurological symptoms (lumbar puncture, electroneurography or biopsy). Lastly, neurological involvement (CNS or PNS) was confirmed in four patients (4.8%) (see Figure 2).

**Figure 3 diagnostics-12-00067-f003:**
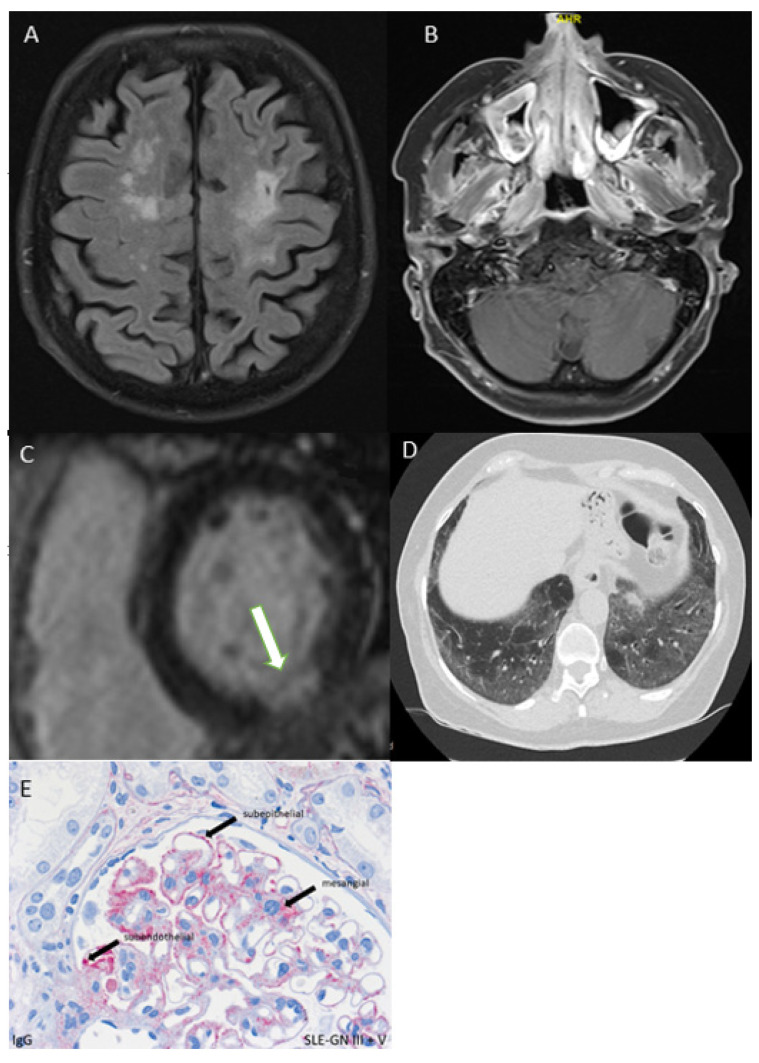
(**A**) Inflammatory involvement of central nervous system with hyperintense lesion in patient with microscopic polyangiitis (MPA); (**B**) mucosal swelling inflammatory swelling of right sinus maxillaris in granulomatosis with polyangiitis (GPA); (**C**) cardiac magnetic resonance imaging (MRI) with late enhancement (arrow), reflecting active cardiac involvement in systemic lupus erythematosus (SLE); (**D**) patient with the onset of SLE without pulmonary symptoms and DLCO of 63 mmol/(min*kPa) and pulmonary ground glass opacity in high-resolution computed tomography (HRCT); and (**E**) renal biopsy presenting lupus nephritis stages III and V.

## 4. Discussion

IRD are potentially associated with organ manifestations. However, data of organ involvement at the onset of IRD are very rare. Therefore, the aim of the present study was to evaluate organ involvement in IRD and the use of a specially developed screening algorithm. 

Our study encompassed 72.6%, 16.7%, and 10.7% of patients with CTD, small-vessel vasculitis, and myositis, respectively. This is in accordance with the modified previous published studies of Hoffmann et al., and Westhoff et al. showing a distribution of CTD of 74.6–86.5%, vasculitis of 6.5–14.4%, and myositis of 7.0–11.0% [15,17]. These percentages refer to the distribution of diseases involved in our study.

There are only very few data on the number of affected organs at the time of IRD diagnosis. This is based, among other things, on the fact that studies on SSc, for example, only focus on pulmonary involvement and not, for example, on additional renal involvement. Additionally, fewer than 50% of patients receive a basic organ screening at onset of IRD [16]. 

### 4.1. Organ Manifestation in IRD

The present study showed a **lung involvement** of 30.8% in SSc, 20.0% in Sjögren’s syndrome, and 12.5% in SLE. Similar results were found in the literature: McNearny et al., and Adler et al. reported pulmonary involvement in SSc of 25% to 33% after a disease duration of 2.8 to 8.5 years [18,19]. A review by Kreider and Highland described similar results with 10% to 20% of pulmonary involvement in Sjögren’s syndrome [20]. According to autopsy reports, up to 18% of patients with SLE had disease-associated pulmonary lesions [21]. In our study, 80.0% of GPA, 83.3% of MPA, and 66.7% of EGPA patients showed lung involvement. Our results were in accordance with the study of Reinhold-Keller et al., reporting 57% of GPA patients with lower respiratory tract manifestations at the onset of the disease [22]. Pulmonary involvement occurred in 25% and 57% of MPA and EGPA patients, respectively [23,24]. Regarding myositis, lung manifestations ranged from 32% to 65% [25]. These data are comparable with our study, revealing 44.4% of myositis patients with pulmonary involvement. In summary, the majority of pulmonary involvement is already present at the time of initial diagnosis. 

Most studies defined **cardiac involvement** in vasculitis as cardiomyopathy, cardiac failure, cardiac arrhythmia, or pericarditis [23,24,26]. Our study revealed cardiac involvement in 11.9% of patients. As shown in the literature, up to 50% of SLE patients have cardiac manifestations (mainly heart valve disease or conduction disturbances) with lower rates for myocardial involvement (9% in clinical trials), which is comparable with our study (6.3%) [27,28,29]. We showed cardiac involvement at initial diagnosis in 3.8% of SSc patients. Autopsy data by Bulkley et al. demonstrated up to 44% of SSc patients with myocardial manifestations, whereas in a clinical trial by Moinzadeh et al., 9% to 18% had cardiac manifestations (defined by heart palpitation, conduction disturbance, diastolic dysfunction) over approximately 18 years of disease duration [30,31]. Our study presented a cardiac involvement of 20.0%, 33.3%, and 100.0% in GPA, MPA, and EGPA patients, respectively. The literature reported cardiac manifestation in 8% to 16% of GPA, 28% of MPA, and 31% of EGPA patients [22,23,24]. A systematic review of Zhang et al. described echocardiographic abnormalities in 14% to 62% of patients with polymyositis or dermatomyositis, which is similar to our results (33.3% cardiac involvement) [32]. 

**Renal involvement** is common in IRD, especially in patients with CTD or vasculitis. In total, our study revealed a renal manifestation in 8.3%. 6.3% of patients with SLE and 7.7% of patients with SSc, which is also in accordance with the literature. In SLE patients, renal involvement is mostly seen in certain ethnic groups [33]. According to Bastian et al., 14% of Caucasian SLE patients had renal manifestations at initial diagnosis [34]. In SSc, renal involvement was described with 8% to 14% [31]. As seen in the literature, up to 27% of EGPA patients show renal abnormalities [24,35]. In GPA and MPA patients, renal manifestations of 51% to 58% and 79% to 100% were described, respectively [22,23,36,37]. In our study, the incidence of vasculitis patients with renal involvement was thus lower than that in the literature (EGPA, 0%; GPA, 20.0%; and MPA, 33.3%). 

Regarding **ENT and GIT involvement**, Anderson et al. described 75% of GPA patients with an ENT manifestations [36], which is comparable to our results (60% ENT involvement). 

In our study, 11.5% of SSc patients presented with GIT involvement at initial diagnosis. According to Moinzadeh et al., the frequency of GIT manifestations increases during the course of the disease (55% of patients with GIT involvement after a disease duration of 7.6 years) [31]. 

### 4.2. Systematic Organ Screening in IRD

In our study, 46.5% of IRD patients had the involvement of one or more organs at the time of diagnosis. This finding underlines the need for a structured organ screening in IRD. Due to the lack of studies on this question, the level of evidence concerning the required diagnostic procedures is low [38,39,40]. To close this gap, we developed a diagnostic algorithm for patients with IRD [29,38,39,41,42,43,44].

Pulmonary screening was the most effective in identifying pulmonary involvement with a combination of two or more lung function parameters, including DLCO (<80%) [45]. Available evidence favours initial PFT as the best surrogate marker screening test for ILD [40,45]. Nevertheless, HRCT is currently the gold standard in diagnosis of lung involvement in IRD [40,46,47]. Pulmonary HRCT appears promising, but radiation is an issue, and the wider use of HRCT could lead to more false-positive results with reduced specificity. In addition, HRCT is usually less available. 

For cardiac involvement, our study revealed that a combination of diagnostic procedures is necessary to screen IRD patients for subclinical cardiac disease (echocardiography, cTnI, cardiac MRI). So far, there are no data for such a combined screening approach, but it is well-known that cTnI is an appropriate biomarker of primary cardiac involvement in IRD [48,49,50], and cardiac MRI is able to detect myocardial inflammation, subendocardial vasculitis, and fibrosis in early and established IRD, even in the absence of echocardiogram abnormalities [51,52,53,54]. Furthermore, arrhythmia (especially ventricular arrhythmia) in IRD is associated with cardiac inflammation, which can be accurately detected by cardiac MRI [55,56]. A high level of experience is required to evaluate a cardiac MRI [53].

For renal manifestation, comprehensive urine analysis with the quantification of urine protein profile und urine sediment at the onset of the disease is strongly recommended [38,41]. This can be usefully supplemented with abdominal sonography and renal biopsy to verify histological kidney involvement. Renal biopsy can be a useful diagnostic technique to adapt therapeutic strategy (e.g., SLE) with a low complications rate [57,58] In detail, major complications (such as transfusion) occur in 2% to 5% of cases, and minor complications (such as haematuria or hematoma) in 8% to 40% of cases in renal biopsy [57,58]. In this context, imaging techniques (such as ultrasound and MRI) were not able to confirm and differentiate renal involvement in IRD. 

For central nerval manifestation, no recommend diagnostic procedure exists. Further, there is evidence that cerebral MRI can detect vascular and inflammatory lesions in IRD [59,60]. Markousis-Mavrogenis et al. observed that subclinical CNS involvement was frequent in IRD patients with cardiac symptoms and cardiac involvement [61]. Consequently, MRT is the optimal technique for the simultaneous detection of cerebral and cardiac inflammatory involvement in IRD [62]. This emphasises that, in the case of suspected multiorgan involvement, systematic screening is mandatory to decide on an adequate therapy. 

In our study, all patients underwent standardised routine screening at the time of initial IRD diagnosis. As is generally the case in medicine, our screening procedure was stepwise, starting with noninvasive measures, followed by more invasive examinations as we progressed. This approach ensures high sensitivity at the beginning, and high specificity thereafter. Of the IRD patients, 46.4% showed an organ involvement. One- and two-organ involvement was present in 29.7% and 10.7% of patients, respectively. The data highlight the need for accurate and detailed organ screening in order to start therapy according to disease stage, and can be adapted if new organ manifestations occur during the course of the disease. However, in some studies, a singular organ manifestation of IRD is initially described, whereas several organs may be affected during the course of the disease [34,63,64]. 

This study evaluated organ manifestations at the onset of IRD. In this context, further longitudinal studies should be initiated to evaluate the organ screening programme regarding the assessment of the treatment response in IRD with differentiation of active and inactive disease.

A limitation of our study is the monocentric and retrospective design. Therefore, the diagnostic algorithm for organ involvement in IRD should be evaluated in a prospective multicentre study design. A further potential limitation of the comparability of organ involvement within the literature data is based on the fact that rheumatologists do not see all patients with IRD and an organ manifestation; for example, patients with renal involvement of IRD are primarily managed in nephrology. In this context, the strength of rheumatology care is the generalised and multiorgan care of IRD patients.

In summary, systematic organ screening in IRD patients is the basis for the structured and stage-appropriate treatment in IRD. This strategy should find its way into rheumatology practice. In a certain way, it is comparable to oncological diseases in which staging is performed before therapeutic decisions (local, systemic, or no therapy) are reached [65,66,67]. Moreover, at which time intervals organ screening should be repeated in order to detect inactive or active organ involvement must be discussed, which leads to an adjustment of the immunosuppressive therapy.

## 5. Conclusions

IRD are a group of heterogenous chronic systemic autoimmune diseases that often show multiorgan involvement, requiring a systematic diagnostic approach. Our study demonstrated a high rate of organ manifestation of 46.4% in patients with IRD at the time of diagnosis. Often, more than one organ were involved, with the lungs and heart being the most affected sites. With this in mind, it can be assumed that more patients than previously thought have undetected organ manifestations at the time of IRD diagnosis. Therefore, thee early detection of asymptomatic organ involvement using comprehensive, systematic screening is mandatory at the onset of IRD, but also in the case of a suspected relapse, to improve the long-term prognosis of these patients. 

## Data Availability

Datasets used and/or analysed during the current study are available from the corresponding author on reasonable request.

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
