# Peer review of "Organ Manifestation and Systematic Organ Screening at the Onset of Inflammatory Rheumatic Diseases"

_diagnostics, 2021, doi:10.3390/diagnostics12010067_

Round 1

Reviewer 1 Report

very interesting work

Please discuss pro and contra of modalities used and explain that the incidence of organs' involvement depends on the modality used.

Finally, discuss the relevant papers by Mavrogeni et al regarding treatment naive ARD patients and combined brain/heart MRI in ARDs

Author Response

Please see the attachment. Many thanks.

Reviewer 2 Report

This manuscript evaluates organ manifestation in newly diagnosed IRD-patients. It concluded that almost half of the patients presented with a solid organ involvement at initial diagnosis of IRD, which contrasts to what has been described in the literature. This work shows that the detection of asymptomatic organ involvement is mandatory at the onset of IRD. I think it would also be interesting the show the influence of therapies and treatments if the authors decide to follow the patients included in this study, as none of the patients were receiving active treatment. 

Author Response

Please see the attachment. Many thanks.

Round 2

Reviewer 1 Report

Congrats! Great work